# A modified rat model of 8 minutes asphyxial cardiac arrest and cardiopulmonary resuscitation

Xin Liu☯, Yan Li☯, Yinghua Gu, Fa Wang, Biyun Tian, Wenxun Liu, Qingshan Ye ⓘ *

Department of Anesthesiology, People's Hospital of Ningxia Hui Autonomous Region, Ningxia Medical University,Yinchuan, Ningxia Hui Autonomous Region, China

☯ These authors contributed equally to this work.
* yeqingshan@hotmail.com

## Abstract

The animal model of cardiac arrest (CA) and cardiopulmonary resuscitation (CPR) serves as a crucial tool for investigating the pathophysiology and treatment strategies associated with cardiac arrest, however, standardized procedures for such models remain insufficiently established. We aimed to modify and specify the existing rat model of asphyxial CA and CPR while providing an analysis of long-term outcomes. A total of 46 rats were allocated into two groups, sham and CA group. In CA group, cardiac arrest was induced through 8 minutes of hypoxia prior to the administration of CPR. In sham group, only tracheal intubation and vascular catheterization were conducted under isoflurane anesthesia. Key parameters along with arterial blood gas results during modeling were meticulously recorded. After a 2-week postoperative observation period, the survival rate of rats and neurobehavioral changes on days 1, 3, 7, and 14 following resuscitation were assessed. Two weeks later, a pathological examination of brain tissue was conducted to evaluate neuronal damage. Results indicated that the average duration of cardiac arrest in CA group was 292.9 ± 12.5 seconds, with a return of spontaneous circulation rate of 78.95% and a survival rate at day 14 reaching 32%. After a duration of 2 weeks, the neurobehavioral scores of the surviving rats returned to their initial baseline levels; however, pathological examination revealed evidence of neuronal damage. In conclusion, we present a refined protocol for establishing a stable rat model of asphyxial CA and CPR, which may assist researchers in this field in enhancing the success rate of modeling.

## Introduction

Research into cardiac arrest (CA) and cardiopulmonary resuscitation (CPR) remains a critical challenge for healthcare professionals globally. The sudden onset of cardiac arrest and the complexity of resuscitation protocols complicate clinical studies on CPR. Therefore, developing reliable animal models for CPR is essential to provide a solid foundation for future clinical research.

**Data availability statement:** All relevant data are within the manuscript and its Supporting Information files.

**Funding:** This work was supported by the Natural Science Foundation of Ningxia Hui Autonomous Region (Grant No. 2024AAC03530).

**Competing interests:** The authors have declared that no competing interests exist.

Currently, methods for inducing cardiac arrest include electrical defibrillation, asphyxiation, and hyperkalemia [1]. Among these, the asphyxiation-induced cardiac arrest model is characterized by precise tissue and organ damage and high stability [2,3]. It aligns more closely with the pathophysiological changes observed in in-hospital cardiac arrest scenarios and effectively simulates a majority of clinical situations involving cardiac arrest. However, this method often results in low success rates due to the extensive instrumentation required, complex procedural steps, and significant neurological impairment associated with its application.Rats are frequently selected by researchers because their hemodynamic indices during cardiopulmonary resuscitation closely resemble those of humans [4]; they are also cost-effective and readily available. Since Hendrickx first described the rat asphyxiation-induced cardiac arrest model in 1984 [5], numerous studies have documented various rat CA models [6]. However, there exists considerable variability in the definitions of key concepts and operational procedures across these studies. The protocols for model preparation remain ambiguous regarding specific steps and details, coupled with a notable lack of assessments concerning long-term survival outcomes and neurological function post-resuscitation in rats.

Therefore, based on the modeling methods of our peers [7] and our prior experimental experience, we refined the procedures for creating a rat model of cardiac arrest induced by 8 minutes of asphyxia. This model demonstrates stable tissue damage and high reproducibility. We also evaluated the long-term prognosis of the rats post-modeling.We aim to offer valuable modeling experience to researchers in the field of cardiopulmonary resuscitation, particularly for early-career experimentalists.

## Materials and Methods

### Animals

The experiment was approved by the Ethics Committee of the People's Hospital of Ningxia Hui Autonomous Region affiliated to Ningxia Medical University (Approval NO. 2024-GZR-10). 46 male Sprague-Dawley rats, weighing 250-300g, were provided by the Animal Center of Ningxia Medical University.Rats were maintained under standard conditions and had ad libitum access to food and water. Room temperature was controlled at 18–24°C, humidity was controlled at 40% to 50%, and a 12h/12h day/night cycle was performed.

### Experimental groups

The animals were divided into two groups by random number method: (1) Sham group (intubation and vascular catheterization under anesthesia without asphyxiation, n=6); (2) CA group (cardiopulmonary resuscitation after asphyxiation-induced cardiac arrest, n=40).

### Anesthesia and intubation

All rats were fasted for 12 hours prior to surgery, but were allowed to drink water freely. The rats were weighed and then placed in an anesthesia box. They were

induced with isoflurane (4%-5% i.a.;RWD,China) inhalation for 3 minutes, and their limbs were fixed in the prone position on the operating table. A visual laryngoscope (MS500,Teslong,China) was used to insert a guidewire 7–8 cm into the trachea through the mouth. A 14G catheter (Insyte BD Medical, Sandy, UT) was then passed through the guidewire to a depth of 4–5 cm (Fig 1). The End-tidal carbon dioxide (ETCO$_2$) monitor (KMI605C,Kingst,China) was connected to the tracheal tube using a three-way connector. The sign of successful intubation was the appearance of a continuous, regular, spike-shaped ETCO$_2$ waveform. After successful intubation, the respirator and anesthesia machine tubing (ABS,Yuyan,China) were connected, and 2% isoflurane was inhaled to maintain anesthesia. An oxygen generator (YU300,Yueyue,China) was connected to the respirator to maintain FiO$_2$ at 1.0. Mechanical ventilation was not initiated immediately, and the rats were allowed to breathe spontaneously.

### Arterial and venous cannula

The right inguinal area of the rats was shaved and disinfected. Lidocaine (1%,1mg/kg l.a.;Zhaohui,Shanghai,China) was used for local infiltration anesthesia of the surgical area skin and subcutaneous tissue. The femoral artery and vein were then exposed layer by layer and a 24G cannula (383083-Y,BD Intima II Plus,China) filled with heparin saline was inserted into each vessel. After successful catheterization, the femoral artery was connected to a biological signal acquisition device (MD-3000D,Zhenghua,China) for monitoring arterial blood pressure (ABP). Simultaneously, the femoral vein was linked to an infusion pump that continuously administered normal saline at a rate of 2 ml/h through the connecting tubing. The cannulation was considered successful when there was backflow in the cannula and a rapid pulsatile flow, and an arterial pressure waveform appeared on the monitor screen. If the rat experienced unexpected minor bleeding during the operation and the hemodynamics showed fluctuations, a small amount of saline solution was slowly infused through the femoral vein to maintain stable blood pressure and heart rate.If the bleeding during the operation was ≥2ml and the vital signs were difficult to maintain, the rats were withdrawn from the experiment.

### Asphyxia induced cardiac arrest

Connect electrodes to the rat's bilateral upper limbs and left lower limb, record the electrocardiogram and heart rate(HR), lubricate the digital rectal thermometer (TH212,Haichuang,China) with petroleum jelly and insert it into the rectum, continuously monitoring rectal temperature (RT) (Fig 2). Rapidly inject cisatracurium besilate (2mg/Kg i.v.;Jianyou,Nanjing,China) through the femoral vein, and once the rat's spontaneous breathing has ceased, turn on the small animal ventilator (DW-3000H,Zhenghua,China) in mechanical ventilation mode, setting the tidal volume to 10 ml/kg, the respiratory

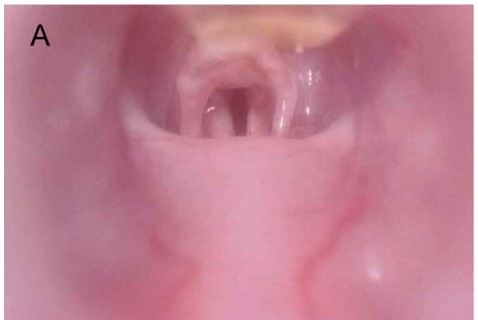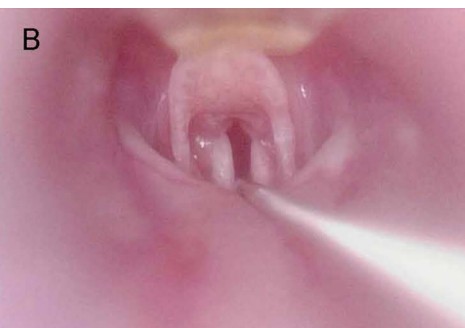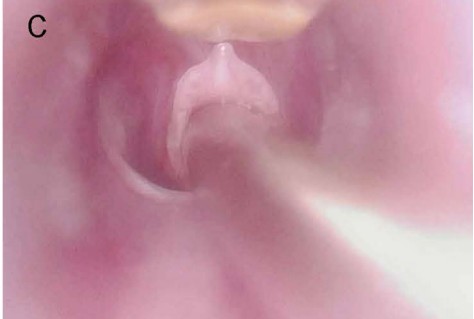

**Fig 1. Endotracheal intubation in rats under the guidance of a visual laryngoscope. (A)** Exposure of rat glottis. **(B)** Placement of a guide wire through the glottis. **(C)** Placement of an endotracheal tube through a guidewire.

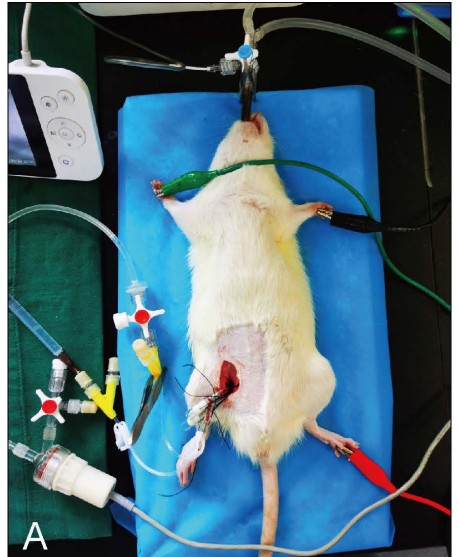
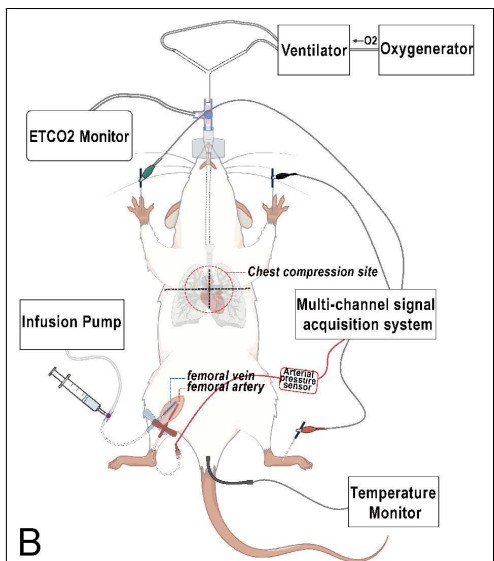
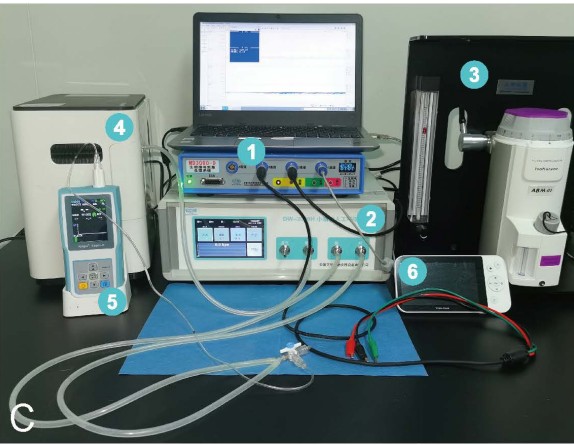

**Fig 2. Schematic overview of rat asphyxial cardiac arrest and cardiopulmonary resuscitation model. (A)** The rat during the molding process. **(B)** Overall diagram of the model. **(C)** Model instruments: ①multi-channel signal acquisition system; ②animal ventilator; ③anaesthesia machine; ④oxygenerator; ⑤ETCO$_2$ monitor; ⑥visual laryngoscope.

frequency to 80 breaths per minute, and the inspiration-expiration ratio to 1:1. The respiratory frequency was subsequently modified based on ETCO$_2$ monitor readings to maintain ETCO$_2$ levels between 30–40 mmHg.

Vital signs were continuously recorded throughout the procedure. After achieving baseline stability for 5 minutes, mechanical ventilation was paused and the tracheal tube three-way valve was closed, initiating an 8-minute countdown. During asphyxia onset, changes in pupil color from red to white were observed; mucous membranes in the oral cavity and nasal area, along with skin around all four paws and inguinal incision tissue transitioned from pink to cyanotic; rectal temperature exhibited a gradual decline; arterial blood pressure initially increased compensatorily before progressively decreasing; ECG recordings displayed various arrhythmias including tachycardia, bradycardia, and premature beats that ultimately led to ventricular fibrillation, mechanical dissociation, and cessation of electrical activity (Fig 3). A mean arterial pressure (MAP) threshold of ≤ 20 mmHg served as criteria for determining cardiac arrest. The time from MAP ≤ 20 mmHg to the end of 8 minutes was recorded as the duration of CA.

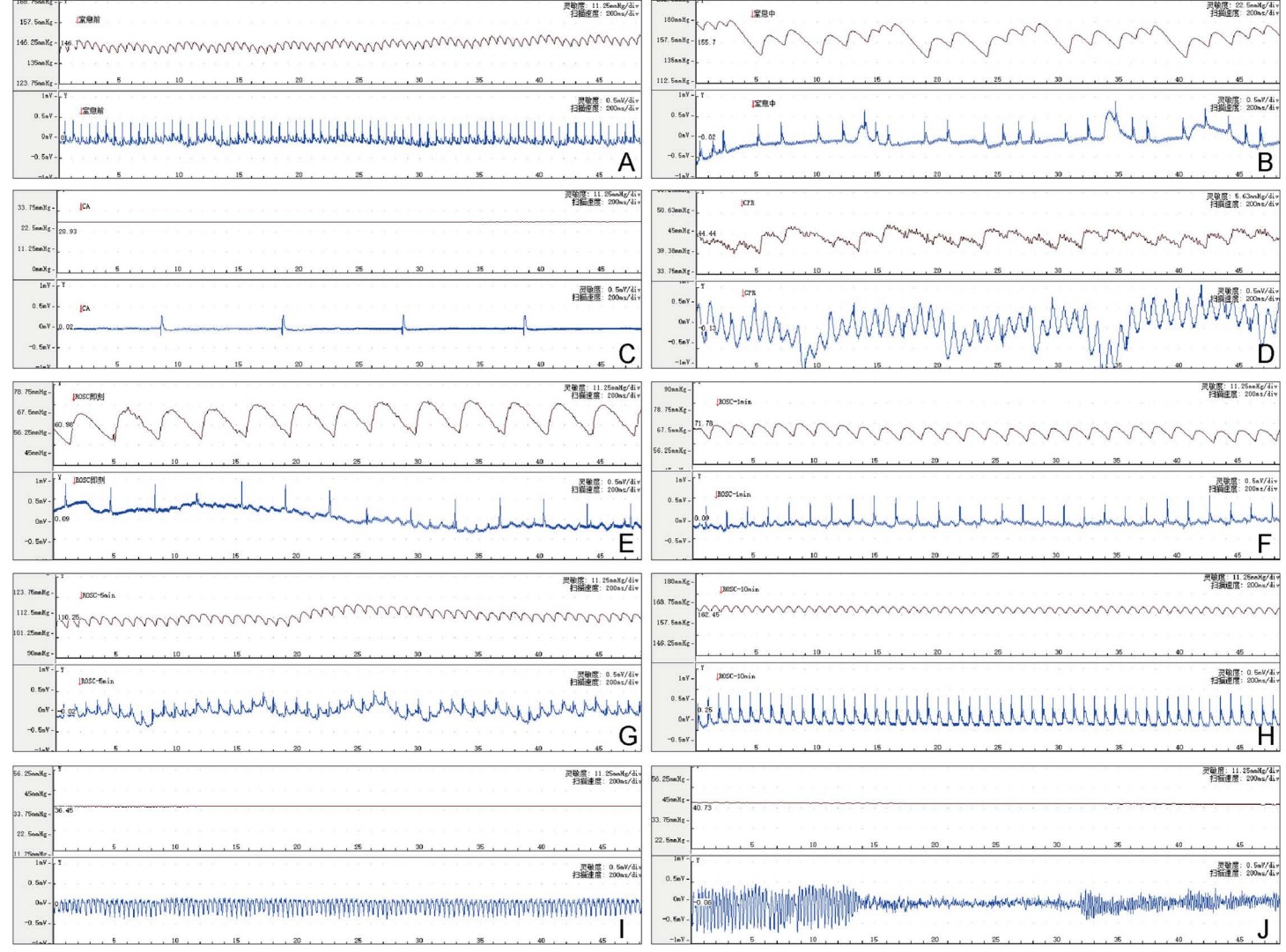

**Fig 3. Typical arterial blood pressure and electrocardiogram during rat modeling. (A)** Before asphyxia; **(B)** During asphyxia; **(C)** Cardiac arrest; **(D)** Cardiopulmonary resuscitation; **(E)**ROSC; (F)1 minute after ROSC; (G)5 minutes after ROSC; (H)10 minutes after ROSC; **(I)**Torsades de pointes; **(J)**Fibrillation. ROSC: return of spontaneous circulation.

## Cardiopulmonary resuscitation

After 8 minutes of asphyxia, the mechanical ventilation mode with a $FiO_2$ of 1.0 was initiated, and the tidal volume was set to 10 ml/kg, the respiratory rate was 80 times/min, and chest compression was immediately started. The single-hand three-finger method (thumb, index finger, and middle finger) was used for chest compression, and the other hand was used to assist fixing the rat's armpits at the double sides. The compression site was the midpoint of the chest wall where the armpits were symmetrically located, and the compression frequency was 200–250 times/min, and the depth was one-third of the anterior-posterior diameter of the chest wall. After chest compression was initiated, epinephrine (0.03mg/Kg) was given intravenously for the first time after 30s of CPR, and then 0.01mg-0.03mg/Kg was given intermittently if necessary. CPR was continued until the return of spontaneous circulation (ROSC), which was defined as a normal

QRS complex on ECG and an MAP ≥ 60 mmHg. Successful resuscitation was defined as MAP ≥ 60 mmHg, duration > 10 minutes, and gradual recovery of redness of the skin and mucosa of the lips and limbs. If ROSC was not achieved within 5 minutes or MAP ≥ 60 mmHg could not be maintained for more than 10 minutes, it was considered a failure of resuscitation.

## Post-resuscitation management

After successful resuscitation, a variable temperature blanket (small-type,Ruyun,China) was used to gradually raise the rectal temperature to 36–37°C. Mechanical ventilation was continued with pure oxygen, and the rats were disconnected from the ventilator when they regained spontaneous breathing (with a respiratory rate of 60–120 breaths/min) and exhibited a gag reflex. Removed the secretions from the trachea and mouth of the rats, and allowed them to breathe air for 5 minutes.If there were no significant fluctuations in vital signs, the endotracheal tube should be extubated. Recorded the duration of intubation after ROSC (from ROSC until extubation). If the rat was unable to resume normal spontaneous breathing after 1 hour of mechanical ventilation, it is considered a failed extubation. In this study, the successful extubation after CPR was taken as the standard for successful modeling of asphyxial cardiac arrest and resuscitation. The femoral arterial and venous catheters were removed and the vessels were ligated, and the incision was sutured in layers and disinfected with iodine. 10 ml/kg of saline was injected into the abdomen to prevent dehydration. After the rats regained consciousness and were able to move independently, they were placed back in their cages and provided with easily accessible food and water. The daily survival and body weight of rats were recorded after modeling. After 2 weeks, the surviving rats were euthanized and the brain tissues were collected for pathological examination. Fig 4 presents a flowchart of the experimental procedure, incorporating a temporal axis.

## Vital signs parameters and arterial blood gas analysis

HR,MAP,$ETCO_2$ and RT were recorded at 6 distinct time points: pre-asphyxia (T0), CA immediately (T1), ROSC immediately (T2), 1 minute after ROSC (T3), 5 minute after ROSC (T4), 10 minute after ROSC (T5), before extubation (T6).

In the CA group, arterial blood samples were collected at three time points: pre-asphyxia, 1 minute after ROSC, and before extubation. Arterial blood gas (ABG) analysis was performed using the blood gas analyzer (GEM Premier 3000,Werfen,Spain). The volume of all blood samples was approximately 200ul, and 200ul of saline was administered to the rats after each sample collection for fluid replacement. No additional drug treatment was given to the rats based on the results of ABG analysis.

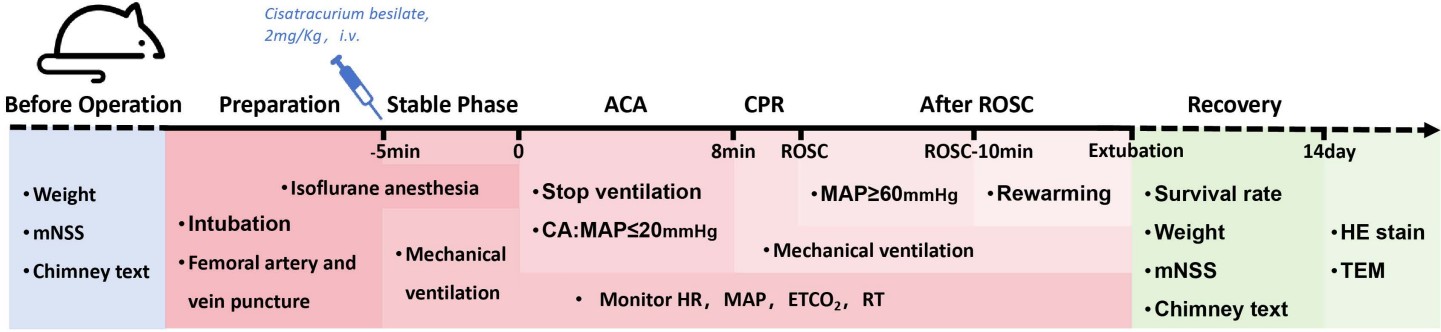

**Fig 4. Experimental flow chart.**

## Neurological behavior tests

On days before asphyxia and 1, 3, 7, and 14 days after resuscitation, two researchers who were blinded to the experimental design evaluated the neurobehavioral behavior of the rats.

The modified neurological severity scoring (mNSS) [8] was used to assess the degree of neurological function damage. The mNSS scale consists of five parts: the tail suspension test (3 points), the straight walking test (3 points), the sensory test (2 points), the balance beam test (6 points), and the loss of reflexes and abnormal movements (4 points), a total of 18 points. Normal SD rats were scored 0 points, 1–6 points were considered mild neurological function damage, 7–12 points were considered moderate neurological function damage, and 13–18 points were considered severe neurological function damage.

The degree of impairment in motor coordination was assessed using the Chimney test [9].The rats were placed upside down at the bottom of a cylinder with an inner diameter of 5.7 cm and a length of 45 cm, and this action would trigger the rat's escape response, causing it to back out of the top of the cylinder. The time from the bottom to the rat's exit from the top of the cylinder was recorded. Normal rats could usually escape successfully within 10–20 seconds, and > 60 seconds was considered a motor function impairment. If a rat failed to exit the cylinder within 120 seconds, the experiment was terminated and recorded as 120 seconds. The evaluation was repeated three times at each time point for each rat, and the shortest escape time was recorded.

## Histopathological examination

Two weeks after establishing the model, brain tissues from the Sham and CA groups were collected for histological examination following PBS perfusion in surviving rats. Half of the brain tissue specimens were fixed with 4% paraformaldehyde, dehydrated with ethanol, embedded in paraffin, and sectioned into 5-micron-thick slices.The slices were stained with hematoxylin and eosin (HE) and observed under a light microscope to assess the degree of brain tissue damage. Each area of the slice was examined in three fields of view, magnified at 40X.The other half of the hippocampal tissue was immersed in precooled 3% glutaraldehyde for over 4 hours for fixation, followed by osmium tetroxide fixation, dehydration, embedding, and preparation of copper grids. The ultrastructure of hippocampal neurons was then observed using a transmission electron microscope (HT-7800, Hitachi, Japan).

## Statistical analysis

Data are presented as mean ± standard deviations (SD). Vital signs and ABG analyses at various time points, along with modeling metrics (weight, intubation duration, and bleeding volume) across multiple groups, were evaluated using one-way ANOVA and Kruskal-Wallis test, followed by Tukey's and Dunn's post hoc tests as appropriate. The parameters (CA duration, CPR duration, and epinephrine) between the two groups were analyzed using unpaired t-tests and Mann-Whitney U tests. The difference of survival rate was analyzed by Kaplan-Meier method. Weight change, neurological scores and behavioral indicators were analyzed by a mixed model of two-way ANOVA with Sidak's post hoc test. Graph-Pad Prism 10 software was used for statistical analysis, and $P < 0.05$ was considered statistically significant.

## Result

### Animals

A total of 46 rats were utilized in the experiment (Fig 5). 40 rats in CA group, with 2 excluded due to unexpected bleeding over 2 ml during the modeling process. As a result, 38 rats experienced asphyxial cardiac arrest,with an average CA duration of 292.9±12.5 seconds. 30 rats successfully achieved ROSC after cardiopulmonary resuscitation, while 8 failed, with a ROSC rate of 78.95% (30/38). Two rats achieved ROSC but failed to wean off ventilation, with the failure reasons being acute respiratory distress in one case and severe pulmonary edema with massive foamy sputum leading to airway

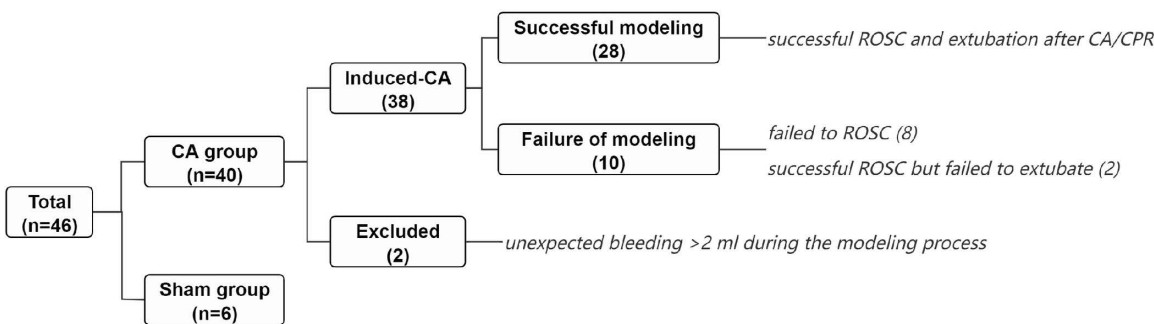

**Fig 5. Groups with number of rats utilized and rationale for exclusion.** The rats in CA and Sham group were observed until the 14th day post-modeling.The surviving rats underwent histopathological evaluation of the brain.

**Table 1. Key metrics of the modeling process.**

|  | CA group | | Sham group (n=6) |
| --- | --- | --- | --- |
|  | Successful modeling(n=28) | Failure of modeling(n=10) |  |
| Weight(g) | 273.3 ± 16.0 | 270.5 ± 15.1 | 273.0 ± 6.5 |
| Intubation duration(s) | 19.6 ± 5.7 | 20.3 ± 6.2 | 22.3 ± 5.8 |
| CA duration(s) | 290.4 ± 11.3 | 300.2 ± 13.5 * | – |
| CPR duration(s) | 104.9 ± 23.5 | 206.6 ± 88.2 *** | – |
| Time from ROSC to extubation(min) | 35.0 ± 8.3 | – | – |
| Bleeding volume(ml) | 0.21 ± 0.40 | 0.25 ± 0.35 | 0.17 ± 0.26 |
| Epinephrine(ug) | 14.6 ± 4.12 | 33.6 ± 7.92 *** | – |

Intubation duration: time from glottis exposure by visual laryngoscope to successful tracheal tube placement. Bleeding volume: the complete infiltration of blood into one cotton ball is approximately equivalent to 1ml of bleeding. *$p$ <0.05, ***$p$ <0.001 compared with successful modeling group.

obstruction in the other. The total number of successfully extubated rats was 28, with a success rate of 73.68% (28/38). All subjects in the sham group were smoothly extubated without any reported complications.

Rats in the failed modeling group experienced longer CA durations than those in the successful group. The prolonged CPR duration also resulted in increased utilization of epinephrine (Table 1).

### Vital signs parameters

Fig 6A. D illustrate the changes in vital signs of the 28 rats that underwent successful modeling during the pre-asphyxia-arrest-resuscitation process. Physiological parameters were not collected during the cardiac arrest phase due to the inability to accurately measure ETCO2 and MAP.Starting from 5 minutes after ROSC, various vital signs gradually returned to their baseline values.HR and ABP were integrated and time-axis compressed by the biosignal acquisition system, presenting the hemodynamic trend map of the entire process (Figs 6E-F).

### Variation trend of ABG

After 8 minutes of hypoxia, 28 rats that successfully resuscitated and were weaned from the ventilator showed significant acidosis changes in their arterial blood gas analysis (Table 2). Without pharmacological intervention to correct the internal environment derangement, sufficient oxygen supply can gradually rectify tissue's acidosis.Compared to pre-asphyxia, there was a stress-induced increase in blood glucose after ROSC, which gradually decreased with prolonged resuscitation

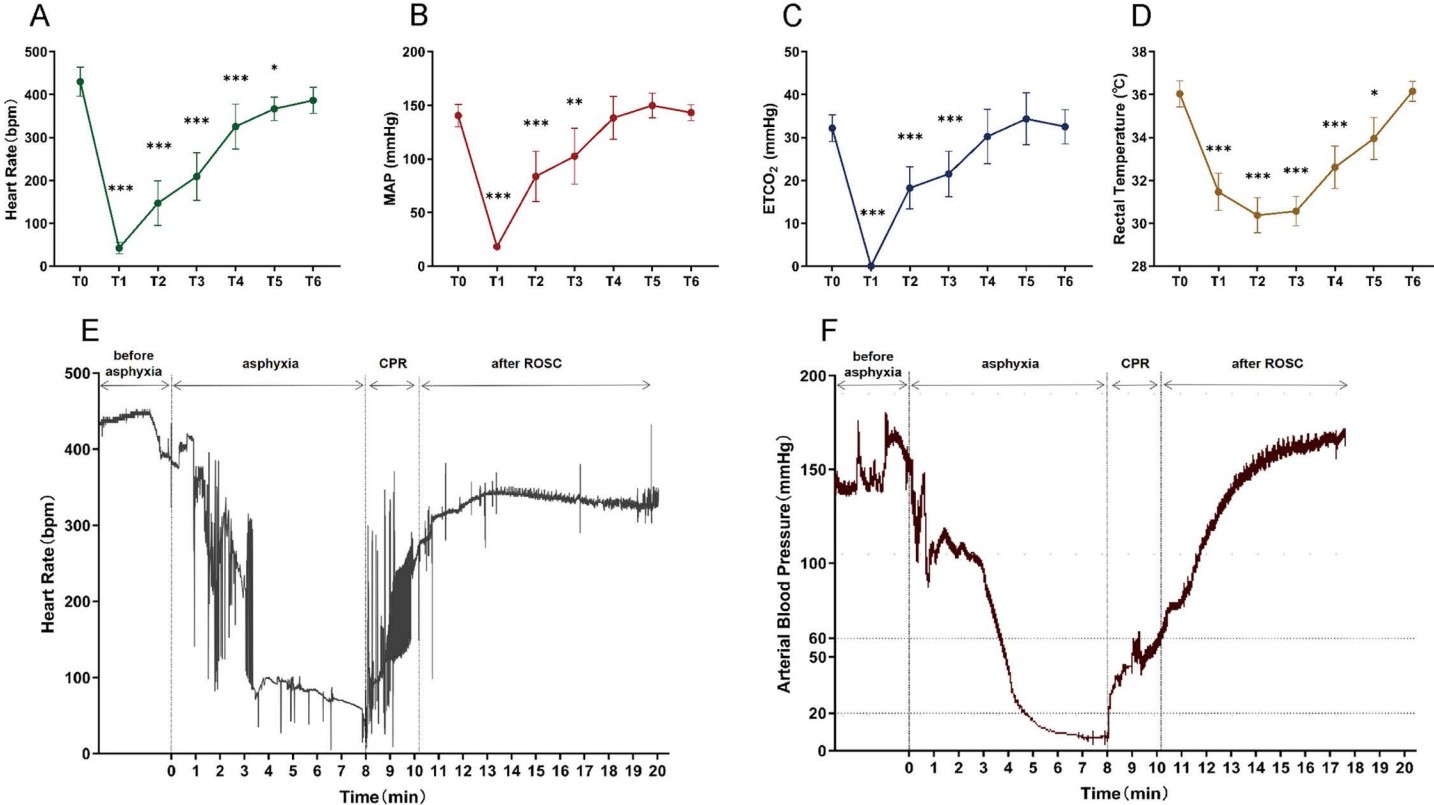

**Fig 6. The variation trend of vital signs parameters during the overall process. (A)** HR; **(B)** MAP; **(C)** ETCO$_2$; **(D)** RT; **(E)** HR trendgram; **(F)** ABP trendgram. T0: pre-asphyxia; T1: CA immediately; T2: ROSC immediately; T3: 1 minute after ROSC; T4: 5 minute after ROSC; T5: 10 minute after ROSC; T6: before extubation. **$p < 0.01$,***$p < 0.001$ at T1-T6 compared with T0.

**Table 2. Arterial blood gas parameters.**

| | Pre-asphyxia | 1 min after ROSC | Before extubation |
|---|---|---|---|
| PH (arterial) | 7.30 ± 0.05 | 6.87 ± 0.05 *** | 7.10 ± 0.08 *** |
| PaO$_2$ (mmHg) | 140.5 ± 48.0 | 72.6 ± 12.0 *** | 134.0 ± 20.0 |
| PaCO$_2$ (mmHg) | 49.0 ± 8.1 | 75.0 ± 17.3 *** | 54.1 ± 6.5 |
| HCO$_3^-$ (mmol/L) | 24.08 ± 2.45 | 11.15 ± 2.93 *** | 17.04 ± 3.14 *** |
| BE (mmol/L) | -1.66 ± 2.77 | -21.55± 3.08 *** | -12.54± 3.81 *** |
| THbc (g/dL) | 11.69 ± 1.32 | 12.93 ± 1.45 ** | 11.84 ± 1.49 |
| Hct (%) | 37.8 ± 4.1 | 41.6 ± 4.7 ** | 38.4 ± 4.5 |
| Glucose (mg/dL) | 195.4 ± 30.6 | 301.1 ± 34.7 *** | 223.5 ± 39.8 * |
| SO$_2$ (%) | 97.7 ± 1.4 | 69.7 ± 11.5 *** | 97.0 ± 0.9 |

ROSC, return of spontaneous circulation; PaO$_2$, partial pressure of oxygen; PaCO$_2$, Partial pressure of carbon dioxide; HCO$_3^-$, bicarbonate ion; BE, base excess; THbc, total hemoglobin; Hct, hematocrit; SO$_2$, saturation oxygen. **$p < 0.01$,***$p < 0.001$ compared with Pre-asphyxia.

time.It is worth noting that although blood was drawn and fluid was infused to dilute the blood after resuscitation, there was no decrease in hematocrit.

## Long-term prognosis after cardiopulmonary resuscitation

A significant mortality rate was observed in rats within the first three days after ROSC, with no further fatalities recorded from day 6 to day 14 (Fig 7A).The survival rate at day 14 after ROSC was determined to be 32% (9/28). During the initial phase after resuscitation, rats exhibited difficulties in feeding and drinking, resulting in progressive weight loss during the first three days;however, by day 7, their weight had returned to baseline levels prior to modeling (Fig 7B). Prior to asphyxia, all rats had an mNSS score of 0 (Fig 7C). The balance beam test indicated a slight alteration in the performance of rats in the Sham group following anesthesia and invasive procedures one day later. The surviving rats in the CA group experienced severe neurological dysfunction for 1 day after ROSC, with the damage gradually improving from moderate to mild between days 3 and 7; by day 14, the neurological function returned to a basically normal state. The chimney test

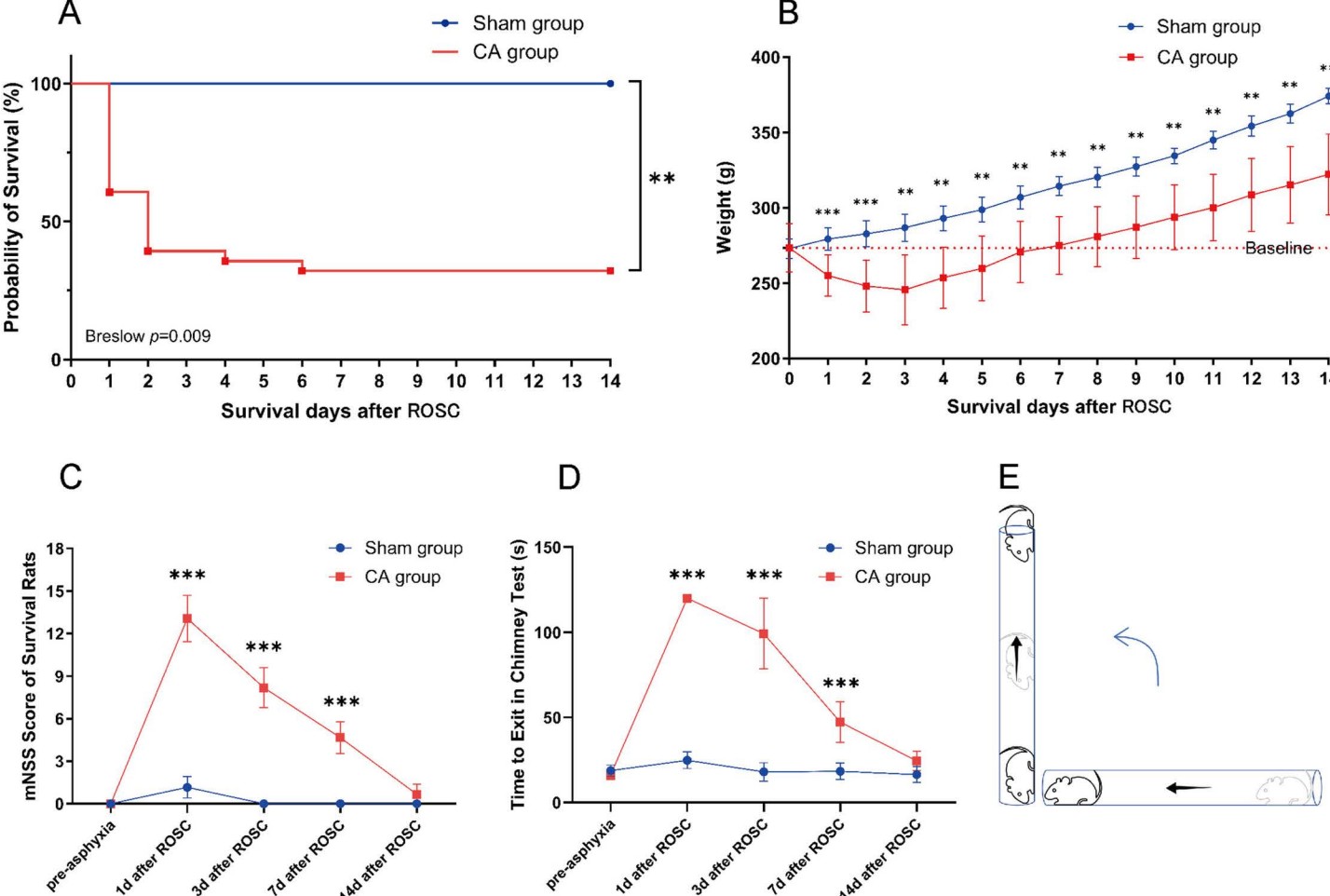

**Fig 7. Long-term outcomes of rats after cardiac arrest. (A)**Kaplan-Meier analyses of cumulative survival during 14-day follow-up after ROSC. Sham group (n=6); CA group (n=28). **(B)**Variations in body weight observed within 14 days after ROSC. **(C)**The mNSS of survived rats at pre-asphyxia, 1,3,7,14 day after ROSC. **(D)**Chimney test results of survived rats at pre-asphyxia, 1,3,7,14 day after ROSC. **(E)**Schematic representation of the operational procedure for the Chimney text. **$p$ <0.01,***$p$ <0.001 compared with Sham group.

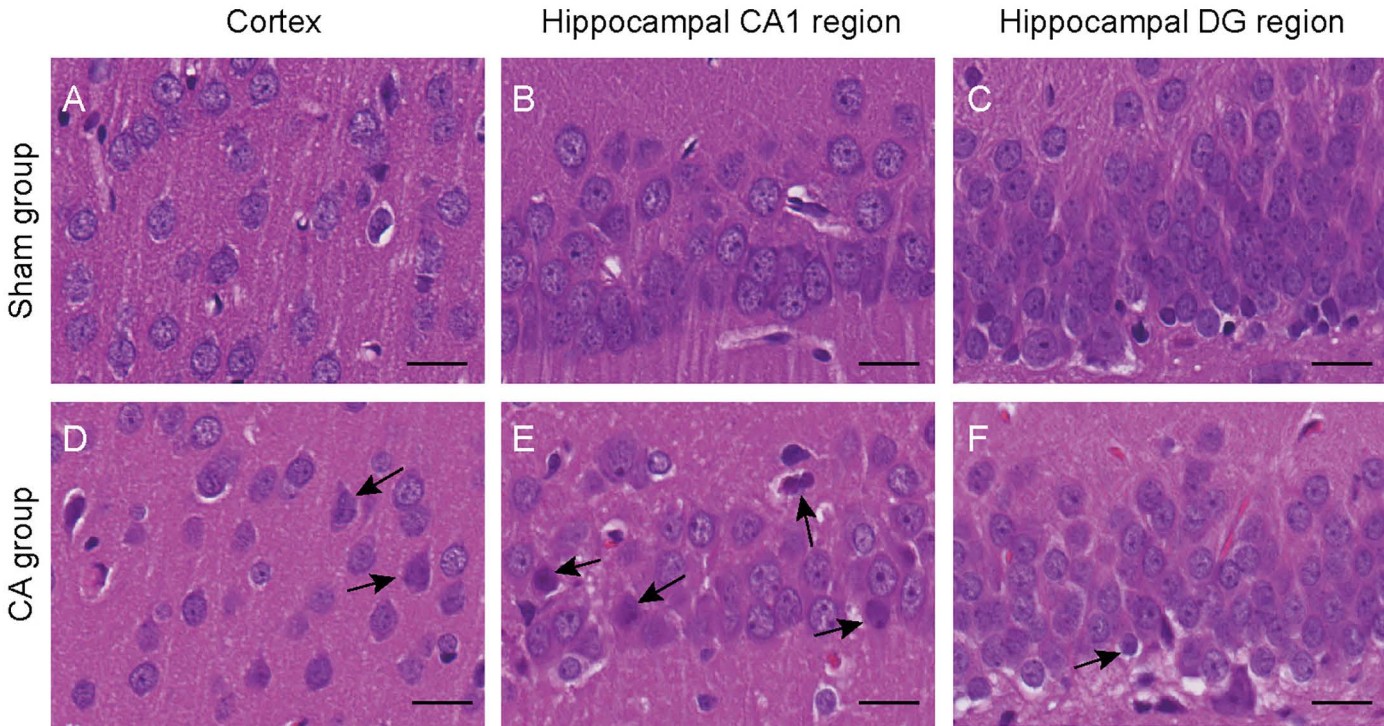

**Fig 8. Neuropathological damage is characterized by alterations observed in HE staining 14 days after resuscitation.** The black arrows in(D),(E),(F) point to damaged neurons.Scale bar=20μm.

elicits the escape response in rats and serves as an indicator of limb motor coordination. Prior to modeling, both groups of rats were able to exit the cylinder quickly (Fig 7D). On day 1, none of the surviving rats in the CA group managed to successfully escape within 120 seconds. Motor function gradually improved over a period of 3–7 days, with all surviving rats demonstrating successful escape within 60 seconds by day 14 after ROSC.

### Cerebral pathological changes observed 2 weeks after resuscitation

Two weeks after CA, brain tissues were harvested from the surviving rats for histological assessment. Compared to the Sham group, rats in the CA group exhibited varying degrees of damage in the cortex, hippocampal CA1 region, and dentate gyrus (DG) area (Fig 8). Particularly, the most severe damage was observed in the hippocampal CA1 region, characterized by disorganized neuronal arrangement, heterogeneous cell sizes, and some cells with condensed nuclei (Fig 8E). Ultrastructural examination of neurons in the CA1 region using transmission electron microscopy revealed that damaged neurons appeared shrunken with irregular nuclear membranes and aggregated heterochromatin (Fig 9C). Additionally, the mitochondrial cristae appeared disordered, fragmented, or dissolved; there was an expansion of the rough endoplasmic reticulum and detachment of ribosomes observed (Fig 9D). Taken together, although the neurological function scores of the surviving rats in the CA group basically recovered to the baseline level after 2 weeks, there were still evident histopathological changes in the brain tissue.

### Discussion

With the widespread adoption of standard CPR, there has been a gradual increase in the rate of spontaneous circulation recovery after CA. However, long-term survival rates and neurological outcomes post-ROSC remain less than optimistic.

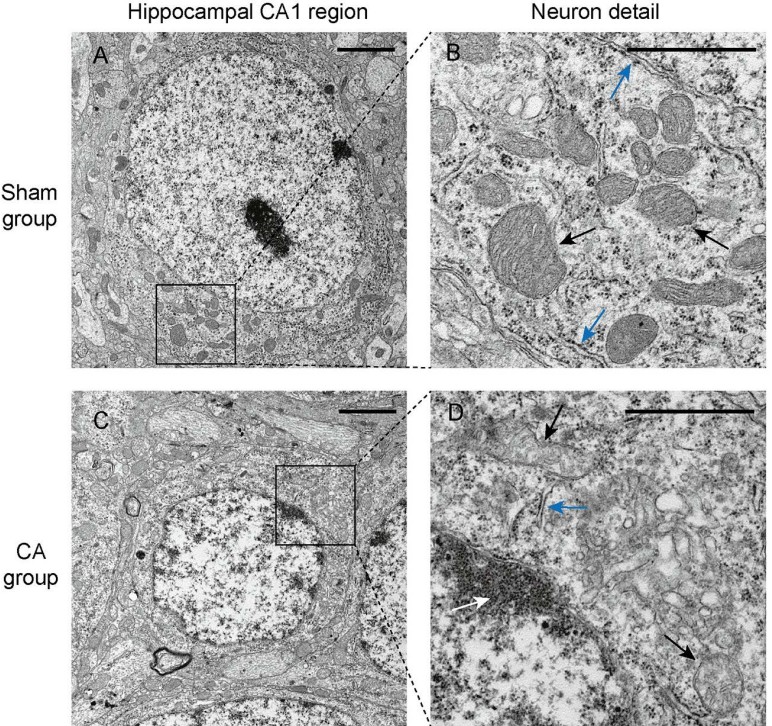

**Fig 9. Neurons in the hippocampal CA1 region 14 days after resuscitation. (A)**Neurons in Sham group.**(B)**Neuronal details in Sham group. Black arrows point to normal mitochondria,blue arrows points to the endoplasmic reticulum, as well as the ribosomes that are attached to it. **(C)** Neurons in CA group. **(D)**Neuronal details in CA group. White arrow indicates the heterochromatin clusters in the nucleus, black arrows indicate mitochondria with disordered, fractured, and dissolved cristae, and blue arrow indicates a decrease in attached ribosomes.Scale bar in **(A)**,(C)=2µm. Scale bar in **(B)**,(D)=1µm.

Since the CA model described in the literature rarely pays attention to the long-term prognosis of animals and generally lacks details of experimental operation, our study provides a detailed description of the complete modeling process for an 8-minutes asphyxial cardiac arrest (ACA) and CPR rat model, along with presenting the long-term prognosis results after resuscitation in rats.

## Methods for inducing cardiac arrest selection

When selecting a method to induce CA, several factors must be taken into account, including the complexity of modeling, the pathophysiological alterations associated with the disease, quality control of the injury, and species-specific characteristics.In previous research, ventricular fibrillation cardiac arrest (VFCA) induced by electrical shock was the most prevalent [1].This preference may arise from the observation that VFCA is associated with higher rates of ROSC [10] in comparison to ACA, and leads to more pronounced myocardial dysfunction [11]. VFCA is favored for investigating non-traumatic out-of-hospital cardiac arrest (OHCA) primarily due to cardiovascular events. However, it is noteworthy that small rodents exhibit rapid baseline heart rates (>200 beats/min), possess robust antiarrhythmic capabilities, and are prone to spontaneous recurrence following cessation of electrical stimulation. Consequently, this results in substantial variability in both the current magnitude and duration required for shock-induced fibrillation to achieve ventricular fibrillation outcomes, thereby impacting model homogenization. Therefore, this results in significant variations in the amplitude and duration of current required to induce VFCA through electric shock, thereby impacting the standardization of the model.

In contrast to OHCA, the etiologies underlying in-hospital cardiac arrest (IHCA) are more intricate, combined cardiopulmonary dysfunctions are frequently observed [12]. Furthermore, prior to the occurrence of cardiac arrest, there is often a preceding phase of organ dysfunction, rendering cardiac arrest as the ultimate outcome rather than an abrupt event. This distinction elucidates why ACA models demonstrate more pronounced multi-organ dysfunction—including severe brain, lung, kidney impairments— and microcirculatory disturbances than VFCA [13,14]. Therefore, in experiments whose main purpose is to investigate the changes in neurological function after CA, the asphyxiation method might be more appropriate. Additionally, total asphyxia duration or CA duration emerges as the primary variable during induction via ACA-facilitating enhanced experimental quality control.

## Details of the key steps for an 8-minute ACA and CPR rat model

In our study, inhalation of isoflurane was selected as the method for anesthesia induction and maintenance, which has become a prevalent approach in recent years for cardiac arrest models [6]. During the induction, we administered 4% to 5% isoflurane for 3 minutes to prevent hypoxic asphyxia resulting from respiratory depression while ensuring an adequate depth of anesthesia. Intubation elicits the most significant stress response among invasive procedures prior to asphyxia, we recommend performing intubation first within 2 minutes after induction, followed by inhalation of 2% isoflurane for maintenance of anesthesia.The question of whether anesthesia should be withdrawn prior to asphyxia remains unclear in most studies. This issue is particularly relevant in model studies utilizing inhalation anesthesia, while intraperitoneal injection anesthesia drugs usually have a longer duration of maintenance for anesthesia. Given that the combination of isoflurane and epinephrine can precipitate severe cardiac arrhythmias [15], it is prudent to withdraw anesthesia before CA is reduced, to mitigate the effects of anesthetics on CPR. In both Hendrickx's [5] and Katz's studies [7], the inhalation anesthetic was flushed for 1–2 minutes, starting 5 minutes before the onset of asphyxia. However, we argue that prematurely discontinuing inhalation anesthesia is not the most appropriate timing. Despite administering muscle relaxants prior to asphyxia, early withdrawal of inhalation anesthesia can still lead to the animal regaining consciousness during the induction of CA, resulting in significant hemodynamic instability [5]. This clearly contravenes ethical standards aimed at maximizing the welfare of experimental animals. In our preliminary investigations, we observed that cessation of isoflurane inhalation via trachea led rats to gradually regain consciousness within 3–5 minutes. Consequently, our experimental protocol involves halting isoflurane inhalation at the onset of asphyxia, ensuring that rats remain anesthetized until just prior to CA, thereby safeguarding animal welfare while minimizing any potential impact of anesthetics on CPR.

Achieving endotracheal intubation within the constrained time frame of inhalation anesthesia presents a challenge for novice experimenters.In the absence of visual laryngoscope, tracheotomy with cervical exposure may be considered as an alternative. While this approach offers clear visualization of the trachea and boasts a high success rate, it is associated with considerable trauma, potential extubation obstruction, and complications during postoperative recovery. Under laryngoscopy (Fig 1A), both the epiglottis and glottis in rats can be clearly visualized, facilitating less traumatic intubation procedures. However, in published papers, investigators did not use a guiding tool to place the catheter directly into the trachea. Currently, various types of needle outer cannulas are primarily employed as substitutes for tracheal tubes; these cannulas are hard with sharp tips that can lead to injuries such as bleeding from epiglottic or vocal cord damage and laryngeal edema when repeatedly directed at the glottis.In our previous exploration, we innovated the use of elastic guide wire as a intubation guide device, which greatly reduced mechanical damage during intubation (Fig 1B-C). With appropriate training, successful intubations were achieved within one minute (Table 1).

After intubation, and in light of the potential lung injury associated with ventilator confrontation [16], we refrained from immediately activating the ventilator for mechanical ventilation. Cisatracurium was administered following the establishment of intravenous access, and ventilation commenced once spontaneous breathing ceased and $ETCO_2$ was monitored. In practice, we observed that fluctuations in $ETCO_2$ values were more sensitive indicators of the final resuscitation outcomes in rats than changes in blood pressure or heart rate, aligning with findings reported in human studies [17].

However, during procedures, it became evident that ETCO$_2$ values among rats with similar body weights and identical mechanical ventilation parameters exhibited significant variability and rapidly deviated from normal ranges. Therefore, it is advisable to select respiratory parameters that maintain ETCO$_2$ within an appropriate range rather than relying on fixed ventilatory settings. These adjusted parameters should be utilized as ventilator settings during CPR. This approach better aligns with the requirements for standardization of physiological indicators among model animals. This approach better aligns with the requirements for standardization.

In recent years, researchers have reached a consensus on the use of muscle relaxants in ACA model [6]. Clamping the tracheal tube while not under muscle relaxation can cause severe struggling, even during anesthesia. If the tracheal tube and arteriovenous catheter are not securely fixed, detachment is likely to occur. Additionally, the outer diameter of the tracheal tube is generally smaller than that of the inner diameter of the trachea. During oral intubation, air may enter lung tissue through both gaps between vocal cords and trachea due to spontaneous breathing in rats which adversely affects asphyxia outcomes.

In ACA model, the duration of asphyxia is the central factor that affects the outcome. There is no unified standard for describing important time periods during asphyxia. Some researchers have recorded the time from the onset of asphyxia to the start of CPR [18,19], which often does not include recording the duration of cardiac arrest. The advantage of this procedure is that the overall time of hypoxia is controllable, and the time recording is accurate and convenient. However, the model with a short total time of hypoxia cannot guarantee enough time of CA for each animal, so that the severity of the model injury cannot meet the experimental requirements.The other method only records the duration of CA, and the standard for CA is usually MAP lower than a certain value [20]. This method can ensure that the time of cardiac arrest in each rat is fixed and the degree of injury is controllable. However, in our experience, if the arterial pressure monitoring device is not equipped with a compression device, although heparin is present in the femoral artery cannula, the catheter tip may still become obstructed, especially when MAP is low. This requires periodic injection of heparin saline into the arterial cannula to maintain a normal waveform. This situation can affect the accuracy of judging whether CA criteria have been met and frequent injection of heparin saline can also impact overall blood volume and coagulation status in rats. Taking all these factors into consideration, we used a total asphyxia time of 8 minutes in our experiment while recording CA duration with MAP < 20 mmHg as the criterion for CA. The results demonstrate that 8 minutes of asphyxia can cause a CA time ranging from 4 to 5 minutes (Table 1), which aligns with "the golden 4 minutes" for human cardiac arrest rescue and achieves an ROSC rate of 79%.

The use of pure oxygen mechanical ventilation (FiO$_2$=1.0) during CPR has been shown to improve the rate of ROSC [21]. However, it is important to acknowledge that different ventilators have varying limitations on gas pressure from external oxygen sources like oxygen bags, cylinders, or oxygenerators. Excessively high pressures can affect the accuracy of tidal volume settings determined by the ventilators and potentially cause pulmonary barotrauma. Therefore, researchers should evaluate how the air source impacts ventilation modes during initial preparations and establish a low-pressure gasbag if needed.

Currently, there is no established standard for the location of chest compressions in rats. Based on our experience, each rat can be marked with a marker pen at the point of strongest heartbeat prior to asphyxia, with a horizontal line drawn at this location,this point generally aligns with the bilateral axillary line of the rat. It is more precise to designate the midpoint of this horizontal line as the site for compression (Fig 2B). In our study, no defibrillation device was employed during cardiopulmonary resuscitation (CPR), and only two rats experienced ventricular fibrillation during resuscitation efforts. While early defibrillation has been shown to enhance rates of ROSC [22], it imposes equipment requirements on researchers.

The administration of epinephrine during CPR is recommended for advanced life support [22], with a commonly used dosage in rat models ranging from 0.005 to 0.04 mg/kg [6]. However, it has been reported that epinephrine is positively correlated with the time required for ROSC and negatively correlated with cardiac function and short-term survival rates

following resuscitation in ACA rat models [23]. It should also be noted that low-dose epinephrine or the absence of epinephrine may prolong the duration of CPR and increase no-flow and hypoperfusion times, potentially adversely affecting long-term survival and neurological outcomes post-resuscitation. In our study, we adjusted the initial dose of epinephrine to 0.03 mg/kg to enhance ROSC rates while minimizing myocardial injury. Furthermore, during our preliminary investigations, we unexpectedly discovered that administering epinephrine in the first 30 seconds after CPR reduced the incidence of acute heart failure and pulmonary edema post-ROSC, compared to administration immediately before or after resuscitation. This finding is consistent with the results of Hendrickx's study, which showed that injecting epinephrine 20 seconds before CPR led to a higher incidence of pulmonary edema [5]. We infer that this effect may be attributed to stagnant circulation exacerbating the side effects of epinephrine. Consequently, we opted to administer a bolus dose of epinephrine 30 seconds post-CPR rather than immediately thereafter [24]. In future studies, we will continue to investigate a more appropriate initial dosage and timing for epinephrine administration in order to further mitigate drug-induced cardiopulmonary injury during resuscitation. During resuscitation, we exclusively utilized epinephrine as the injectable drug, deviating from Katz's protocol [7]. Katz administered intravenous sodium bicarbonate immediately following epinephrine to regulate blood pH, achieving a 100% ROSC rate; however, pre-extubation blood gas measurements were not provided. Our study revealed that, even in the absence of sodium bicarbonate administration, the arterial blood $PaCO_2$ levels in rats exhibited an initial increase followed by a gradual decrease within one hour post-ROSC by adjusting respiratory parameters. We establish an index of the internal environment within one hour post-ROSC under minimal pharmacologic intervention, which can serve as a guideline for optimizing the timing and dosage of sodium bicarbonate administration. However, it is important to acknowledge that withholding sodium bicarbonate may result in reduced rates of ROSC and survival.

Given that many studies require rats to survive for a minimum of 24 hours post-resuscitation, it is crucial to assess whether these animals can be weaned and extubated as promptly as possible following resuscitation. Our study revealed that not all rats with ROSC lasting longer than 10 minutes were successfully weaned and extubated. Specifically, two rats failed to extubate after ROSC (Fig 5), while all rats demonstrating successful resuscitation were intubated within 1 hour post-ROSC (Table 1). We defined the failure of extubation as the inability to restore normal spontaneous breathing offline and the requirement for mechanical ventilation over 1 hour. The rationale for this one-hour timeframe is based on observations that rats unable to resume normal spontaneous breathing often complicated with lung injury, prolonged mechanical ventilation may exacerbate such injuries [25], thereby affecting the overall homogeneity of injury severity in the model. Additionally, extended periods of mechanical ventilation occupy ventilator resources and hinder modeling efficiency. Some researchers intentionally administer prolonged periods of pure oxygen ventilation (>1 hour) to laboratory rats in order to enhance short-term survival rates and improve neurological recovery [7]. Even after spontaneous breathing resumes, muscle relaxants are administered to ensure continuous pure oxygen ventilation, which was called the ICU stage [5]. The advantage of our weaning protocol is to minimize the intervention on physiological changes in respiratory function after resuscitation in rats; however, it may also aggravate neurological damage and reduce short-term survival due to insufficient tissue oxygen supply.

In addition to respiratory support, targeted temperature management (TTM) is a critical component of post-resuscitation care. Hypothermia is recognized for its neuroprotective effects following CPR [26]. Furthermore, research has demonstrated that epinephrine administration during hypothermic cardiopulmonary resuscitation can alleviate cardiac insufficiency caused by epinephrine [27]. In addition to experiments designed to investigate the neuroprotective effects of hypothermia, which will take the target hypothermia as the control purpose to orderly cool and warm the animals, other studies primarily employ incandescent lamps or variable-temperature blankets to maintain body temperatures within normal ranges throughout the modeling process [6]. While this approach of sustaining a constant normal body temperature can mitigate the impact of temperature variations on experimental subjects to some extent, it may inadvertently exacerbate brain injury and compromise resuscitation rates. We continuously monitored rectal temperatures in rats but refrained from using heating devices from the onset of anesthesia until 10 minutes post-ROSC. The results indicated that

body temperatures spontaneously decreased by 4–6°C following asphyxia onset and began to gradually rise after ROSC, with hypothermia (30–35°C) persisting until 10 minutes post-ROSC (Fig 6D). Spontaneous hypothermia lasted approximately 18 minutes from asphyxia onset until we initiated rewarming using a warming device, which positively influenced prognosis.

### Long term prognosis evaluation of an 8 minutes ACA and CPR rat model

The assessment of neurological function has consistently been a focal point in CA models, as brain injury is the primary factor that influences long-term survival rates in patients who experience CA [28]. Perhaps due to the high mortality rate after CPR, many researchers choose to collect samples 24 hours after ROSC. Studies that require early neurological evaluation typically set the endpoint at 72 hours [29], whereas one week after ROSC signifies a crucial point between short-term and long-term results [30]. Our study extends the observation period for prognosis to 14 days, which is relatively uncommon within this model. However, we assert that a longer observation duration post-ROSC holds significant value. Following intervention without any specific pharmacological agents or treatment methods, it is essential to evaluate the reparative capacity of various organ systems in rats. This evaluation can provide multiple baseline states for experimental personnel assessing the effectiveness of subsequent interventions, allowing researchers to select sampling time points based on their needs regarding the severity of neurological function impairment.Additionally, studying long-term survival rates offers valuable insights for peer researchers concerning sample size calculations at different experimental design time points.

In the course of our research, we found that the death of rats primarily occurred within the first three days after resuscitation, and no additional fatalities were observed after seven days (Fig 7A), which correlated with the severity of neurological impairment in these animals. After 14 days, neurological function was largely restored to baseline levels (Fig 7C). Considering the dynamic changes in prognostic indicators, we propose that the optimal time for early evaluation after resuscitation is between 1–3 days, while a period of 7–14 days should be designated for long-term assessment. Although the main signs of rats have basically recovered by day 14 post-resuscitation, neuronal damage can still be observed from the microscopic level of brain tissue (Fig 8 and Fig 9), which may adversely affect long-term life quality.

### Limitations

This study has some potential limitations. Firstly, we only used adult male rats for the experiments, without considering the impact of age and sex differences on model stability. Secondly, we know that most patients experiencing CA present with complications involving the heart, lungs, and brain,while all rats in this study were healthy. To more accurately simulate the clinical scenario, it is essential to incorporate various complications into this model to better reflect clinical realities.

### Conclusion

In summary, we have established a stable 8 minutes asphyxial cardiac arrest and cardiopulmonary resuscitation rat model and assessed the long-term prognosis, providing a foundation for further research into the mechanisms underlying cardiac arrest and potential intervention strategies. We believe that this detailed protocol can serve as an experimental reference for researchers in same field.

### Supporting information

**S1 Data. Supplement SDrat-raw data.** SD rat raw data.
(XLSX)

**S1 Image. Supplement Raw Image-HE.** HE images.
(PDF)

**S2 Image. Supplement Raw Image-TEM.** TEM images.
(PDF)

## Acknowledgments

We sincerely acknowledge the invaluable support and assistance provided by the staff of the Experimental Animal Center at Ningxia Medical University in rat rearing. Our heartfelt gratitude goes to Kerong Hai and Xu Ma for their exceptional expertise and invaluable help in establishing this model. Figure support was provided by Figdraw.

## Author contributions

**Conceptualization:** Yan Li, Wenxun Liu.

**Data curation:** Xin Liu, Yan Li, Yinghua Gu.

**Methodology:** Xin Liu, Yan Li, Yinghua Gu, Fa Wang, Biyun Tian.

**Project administration:** Xin Liu, Qingshan Ye.

**Resources:** Xin Liu, Yan Li, Yinghua Gu, Fa Wang, Qingshan Ye.

**Supervision:** Qingshan Ye.

**Validation:** Xin Liu, Fa Wang, Biyun Tian.

**Visualization:** Xin Liu.

**Writing – original draft:** Xin Liu, Yan Li.

**Writing – review & editing:** Xin Liu, Qingshan Ye.

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
