## [Decision Letter · Decision Letter 0]

14 Jan 2025

PONE-D-24-45627A modified rat model of 8 minutes asphyxial cardiac arrest and cardiopulmonary resuscitationPLOS ONE

Dear Dr. Ye,

Thank you for submitting your manuscript to PLOS ONE. After careful consideration, we feel that it has merit but does not fully meet PLOS ONE’s publication criteria as it currently stands. Therefore, we invite you to submit a revised version of the manuscript that addresses the points raised during the review process.

**ACADEMIC EDITOR: **

Please address minor concerns raised by the reviewer 2

A marked-up copy of your manuscript that highlights changes made to the original version. You should upload this as a separate file labeled 'Revised Manuscript with Track Changes'.An unmarked version of your revised paper without tracked changes. You should upload this as a separate file labeled 'Manuscript'.

We look forward to receiving your revised manuscript.

Kind regards,

Vadim Ten MD, PhD

Academic Editor

PLOS ONE

Journal Requirements:

Additional Editor Comments:

Dear Qingshan Ye.

Your manuscript has been reviewed by two independent reviewers and requires a minor revision to to be published.

Please, address minor concerns of the reviewer 2

Sincerely,

Vadim S. Ten MD, PhD

Reviewers' comments:

Reviewer's Responses to Questions

**Comments to the Author**

1. Is the manuscript technically sound, and do the data support the conclusions?

Reviewer #1: Yes

Reviewer #2: Yes

2. Has the statistical analysis been performed appropriately and rigorously?

Reviewer #1: Yes

Reviewer #2: Yes

3. Have the authors made all data underlying the findings in their manuscript fully available?

Reviewer #1: No

Reviewer #2: Yes

4. Is the manuscript presented in an intelligible fashion and written in standard English?

Reviewer #1: Yes

Reviewer #2: Yes

5. Review Comments to the Author

Reviewer #1: In this manuscript, the Authors endeavored to develop a refined protocol for a rat model of cardiac arrest and cardiopulmonary resuscitation. They argue that the described by them model “aligns closely with the pathophysiological changes observed in in-hospital cardiac arrest scenarios and effectively simulates a majority of clinical situations involving cardiac arrest”. The Authors now provide a very helpful narrative for step-by-step procedures and guidelines for standardization of experimental outcomes.

There are many merits to this manuscript. The study is well planned, experiments are well executed, and the document is expertly written. I concur with the Authors that longitudinal (14 days) evaluation of mortality and neurological outcomes is very helpful and provides peer researchers with a benchmark for sample size calculations at different experimental design time points. The limitations of the work are well described.

Below are several specific suggestions for improvements in data reporting and discussion of the literature:

[1] Moderate concern: As the proposed approach is not original, it would be very helpful to compare this study’s outcomes to the previous reports, both in the context of successful resuscitation and longer-term survival. The original methodological articles by HHL Hendrickx et al. (Resuscitation, 1984) and L Katz et al (JFRBM, 1995) are mentioned in Introduction but largely excluded from Discussion. To the best of this reviewer’s knowledge, Katz and colleagues reported much higher 72-h survival rates while using qualitatively similar procedure (why?). This and other studies deserve additional discussion.

[2] Moderate concern: The supplemental file is missing primary data for neurological outcomes presented in Figure 7C and 7D.

[3] Minor concern and suggestion for improvement: I would suggest including in the Supplemental files additional H&E staining images from more animals and more EM images. Such additional information will provide the specialists with a stronger basis for evaluating neurological damage in ACA brains. There is significant demand for such data for research and educational purposes.

Reviewer #2: The authors in this study established a stable 8 minutes asphyxial cardiac arrest and

cardiopulmonary resuscitation rat model and assessed the long-term prognosis,

providing a foundation for the researchers in the field into the mechanisms underlying cardiac

arrest and potential intervention strategies. This is an elegant study with extensive details on methodology that can serve as an experimental reference for the field. Further limitations of the study has been considered such as the use of only male but not the female rats. Overall, this is a very timely and well conducted study and is viewed as a significant resource for the field.

6. PLOS authors have the option to publish the peer review history of their article (what does this mean? ). If published, this will include your full peer review and any attached files.

**Do you want your identity to be public for this peer review?** For information about this choice, including consent withdrawal, please see our Privacy Policy .

Reviewer #1: No

Reviewer #2: No

---

## [Author Response · Author response to Decision Letter 1]

22 Feb 2025

Dear Editors and Reviewers:

Thank you for the opportunity to revise our manuscript and address the reviewers’ comments. We appreciate the thorough evaluation by the academic editor and reviewers, and we have carefully revised the manuscript to incorporate their suggestions. Below, we provide a point-by-point response to all concerns raised.

Response to Academic Editor’s Comments

Editor’s Comment: [Address minor concerns raised by the reviewer].

Response: We thank the editor for this observation. We have discussed the issues raised by the reviewers in more detail in the revised manuscript and clearly marked the revisions. We supplemented the re-uploaded attachment with more detailed original neurobehavioral data and uploaded additional HE images and TEM images of animals in the PACE system.

Response to Reviewer 1’s Comments

Reviewer 1:

Opinion 1: [Please compare the results of this study with those reported by HHL Hendrickx (Resuscitation, 1984) and L Katz (JFRBM, 1995) in the discussion, and answer why the 72-hour survival rate was higher in the Katz’s study when a similar procedure was used.]

Response: We appreciate this suggestion. In the newly submitted manuscript, we discuss and contrast the original methods presented in the article by Hendrickx and Katz, which are marked in red under "Revised Manuscript with Track Changes." Regarding the difference in the 72-hour survival rate between our experiment and that of Katz, a certain elaboration has also been made in the discussion section of the revised manuscript. I summarize the possible reasons as follows:

①Use of sodium bicarbonate during cardiopulmonary resuscitation. To minimize the effect of drugs on blood gas results after ROSC, sodium bicarbonate was not used in our experiment, which may be one of the reasons for the lower ROSC rate and 72-hour survival rate compared to the Katz study.

②Post-ROSC longer pure oxygen mechanical ventilation. After ROSC, when the spontaneous breathing of the rats recovered to the extubation conditions, we implemented the weaning and extubation protocol, and did not use muscle relaxants to keep the rats on pure oxygen mechanical ventilation continuously. However, Katz's protocol maintained at least 1 hour of pure oxygen ventilation. The rapid extubation protocol without longer ICU monitoring may lead to an increase in the number of rats that died due to respiratory system dysfunction in the early stage of resuscitation.

③The difference in body weight of SD rats leads to different tolerance to hypoxia. We selected rats weighing 250-300g, which were in the early adult stage. Katz selected rats weighing 350-400g. Adult rats with lower body weight and younger age have more active basal metabolism, higher oxygen consumption, faster lactate accumulation during hypoxia, are prone to acidosis, and have poorer tolerance to hypoxia in brain tissues [M J Durkot Aviat Space Environ Med,1986�,M A Holliday(Pediatr Res,1967)].

To sum up, in our experimental results, the ROSC rate and 72-hour survival rate of rats were lower than those observed in Katz's study. However, based on the comprehensive neurological function score, behavior, and pathological results, 72 hours after resuscitation, the neurological function damage in our rats was more severe compared to that in Katz's rats. The neurological function damage in Katz's experiment was mild at 72 hours. We consider that more severe neurological function damage in the early stage of resuscitation may be more representative of the clinical situation.

Opinion 2: [The supplemental file is missing primary data for neurological outcomes presented in Figure 7C and 7D.]

Response: We agree with the reviewer’s concern. We uploaded all the original data of SD rats in the research in the previous attachment, including the data of "mNSS score and Chimney test in Neurobehavioral experiments". This time, we supplemented the scores of each part within the total score of mNSS score in the original data of SD rats. Their locations can be found in the newly submitted Excel file "SDrat - raw data" .

Opinion 3: [It is recommended to supplement additional H&E staining images of more animals and more EM images.]

Response: We appreciate the suggestions of the reviewers. We have supplemented more HE images and TEM images of animals in PACE. For their locations, please refer to the newly submitted supplementary PDF file: "Supplement Raw Image - HE & TEM".

We believe the revised manuscript now addresses all concerns. We are grateful for the reviewers’ time and constructive feedback. Please do not hesitate to contact us if further revisions are required.

Sincerely,

Qingshan Ye

People's Hospital of Ningxia Hui Autonomous Region,Ningxia Medical University

No.301 Zhengyuan North Street, Yinchuan, Ningxia,China,750001

E-mail:yeqingshan@hotmail.com

22/02/2025

---

## [Decision Letter · Decision Letter 1]

23 Mar 2025

A modified rat model of 8 minutes asphyxial cardiac arrest and cardiopulmonary resuscitation

PONE-D-24-45627R1

Dear Dr. Qingshan Ye,

We’re pleased to inform you that your manuscript has been judged scientifically suitable for publication and will be formally accepted for publication once it meets all outstanding technical requirements.

Kind regards,

Vadim Ten

Academic Editor

PLOS ONE

Additional Editor Comments (optional):

Reviewers' comments:

Reviewer's Responses to Questions

**Comments to the Author**

1. If the authors have adequately addressed your comments raised in a previous round of review and you feel that this manuscript is now acceptable for publication, you may indicate that here to bypass the “Comments to the Author” section, enter your conflict of interest statement in the “Confidential to Editor” section, and submit your "Accept" recommendation.

Reviewer #1: All comments have been addressed

2. Is the manuscript technically sound, and do the data support the conclusions?

Reviewer #1: Yes

3. Has the statistical analysis been performed appropriately and rigorously?

Reviewer #1: Yes

4. Have the authors made all data underlying the findings in their manuscript fully available?

Reviewer #1: Yes

5. Is the manuscript presented in an intelligible fashion and written in standard English?

Reviewer #1: Yes

6. Review Comments to the Author

Reviewer #1: The authors have addressed concerns raised during review of the previous version of this manuscript.

7. PLOS authors have the option to publish the peer review history of their article (what does this mean? ). If published, this will include your full peer review and any attached files.

**Do you want your identity to be public for this peer review?** For information about this choice, including consent withdrawal, please see our Privacy Policy .

Reviewer #1: No

---

## [Editor Report · Acceptance letter]

PONE-D-24-45627R1

PLOS ONE

Dear Dr. Ye,

I'm pleased to inform you that your manuscript has been deemed suitable for publication in PLOS ONE. Congratulations! Your manuscript is now being handed over to our production team.

Kind regards,

on behalf of

Professor Vadim Ten

Academic Editor

PLOS ONE